# Recent Advances in Electrochemical Nitrogen Reduction Reaction to Ammonia from the Catalyst to the System

Yong Hyun Moon [1,†], Na Yun Kim [1,†], Sung Min Kim [1] and Youn Jeong Jang [1,2,*]

1 Department of Chemical Engineering, Hanyang University, 222 Wangsimni-ro, Seoul 04763, Korea
2 Institute of Nano Science and Technology, Hanyang University, 222 Wangsimni-ro, Seoul 04763, Korea
* Correspondence: yjang53@hanyang.ac.kr
† These authors contributed equally to this work.

**Abstract:** As energy-related issues increase significantly, interest in ammonia ($NH_3$) and its potential as a new eco-friendly fuel is increasing substantially. Accordingly, many studies have been conducted on electrochemical nitrogen reduction reaction (ENRR), which can produce ammonia in an environmentally friendly manner using nitrogen molecule ($N_2$) and water ($H_2O$) in mild conditions. However, research is still at a standstill, showing low performances in faradaic efficiency (FE) and $NH_3$ production rate due to the competitive reaction and insufficient three-phase boundary (TPB) of $N_2(g)$-catalyst(s)-$H_2O(l)$. Therefore, this review comprehensively describes the main challenges related to the ENRR and examines the strategies of catalyst design and TPB engineering that affect performances. Finally, a direction to further develop ENRR through perspective is provided.

**Keywords:** ammonia; nitrogen reduction reaction; electrocatalyst; system engineering; three-phase boundary





## 1. Introduction

Ammonia ($NH_3$) is an essential commodity for chemicals utilized in various industries, such as fertilizers, plastics, dyes, pharmaceuticals, and so on [1,2]. In particular, its use as a fertilizer made mass production of food possible and led to a population increase. Recently, $NH_3$ has been focused on as a sustainable fuel and a hydrogen storage medium with the promising properties of 5.52 kWh $kg^{-1}$ high energy density and 17.6% high hydrogen content [3]. According to the US Geological Survey, $NH_3$ production in 2021 has gradually increased to as much as 150 million tons [4].

Most $NH_3$ production mainly relies on the traditional Haber–Bosch process which converts hydrogen ($H_2$) and nitrogen ($N_2$) directly into $NH_3$ ($N_2 + 3H_2 \rightleftharpoons 2NH_3$) using Fe-based catalysts [5]. However, since the Haber–Bosch process requires extreme operating conditions—such as a temperature and pressure of over 400 °C and 20 MPa, respectively—to cleave the strong triple-bond energy (941 kJ $mol^{-1}$) of $N_2$ molecules, the process annually consumes a significant amount of energy accounting for around 1–3% of globally produced energy [6–9]. Furthermore, the requirement of the enormous $H_2$ supply generated by fossil fuels leads to significant carbon dioxide ($CO_2$) emissions, which amounted to 300 million tons [8]. When we convert it to an energy requirement per ton of $NH_3$ production, the theoretical energy requirement for the Haber–Bosch process is 22.2–28.8 GJ $t_{NH3}^{-1}$. Furthermore, $CO_2$ emission from the Haber–Bosch process is expected to be 1.673 $t_{CO2}$ $t_{NH3}^{-1}$ [10,11]. Considering the increasing need for $NH_3$ and the Haber–Bosch process' economic and environmental issues, a novel environmentally friendly and economically efficient $NH_3$ production process is in high demand.

The electrochemical nitrogen reduction reaction (ENRR) is one of the most attractive $NH_3$ production processes. $NH_3$ can be produced using $H_2O$ as a hydrogen source and $N_2$ gas at mild temperature and pressure under biased conditions. Theoretical minimum energy requirement for ENRR is expected to be only 19.9 GJ $t_{NH3}^{-1}$, as assuming ideal

1.17 V supplying for overall $NH_3$ production with 100% of faradaic efficiency. Furthermore, a negligible amount of $CO_2$ during ENRR is expected, if electricity supplied is from renewable energy such as wind and solar and the possibility of $CO_2$ emission from process engineering is excluded such as product separation, input gas compression, and so on [10,11]. Thus, ENRR can significantly reduce $CO_2$ emission and energy consumption compared to the Haber–Bosch process. However, ENRR has several critical challenges hindering its practical applications due to the insufficient performances recorded so far. For example, to apply the ENRR as a major $NH_3$ production pathway in industrial systems, an $NH_3$ production rate of at least 6120 µg h$^{-1}$ cm$^{-2}$ should be achieved [12,13]. However, despite great efforts for advancing the ENRR's performance, the $NH_3$ production rate is still significantly low with 612 µg h$^{-1}$ cm$^{-2}$ and a Faradaic efficiency (FE) below 10% [14]. Thus, it is critical to understand the ENRR's major limitations and the previous efforts of overcoming the challenges to propose the next generation of ENRR, which would be a practical available alternative to the Haber–Bosch pathway.

In this review, we will first discuss the essential ENRR thermodynamics fundamentals, kinetics related to reaction mechanisms, and other factors affecting the ENRR's performance. Next, we review the widely studied catalyst design strategies and reaction environmental engineering. Moreover, we will thoroughly investigate each strategy's effect on the ENRR's performance (Scheme 1). Finally, we end this review by showing ENRR's great prospects and the direction for further investigation to apply it in the future.

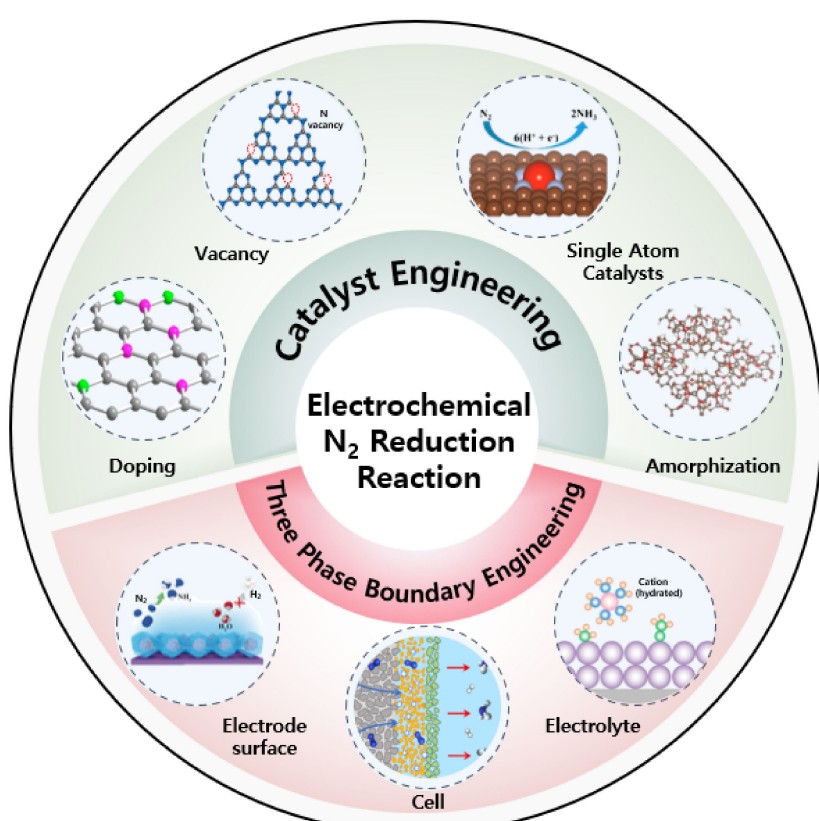

**Scheme 1.** Schematic diagram of electrochemical $N_2$ reduction reaction performance improvement strategies covered in this paper. Schematic of doping, reproduced with permission [15]. Copyright 2021, Elsevier B.V. Schematic of vacancy, reproduced with permission [16]. Copyright 2020, American Chemical Society. Schematic of single-atom catalysts, reproduced with permission [17]. Copyright 2018, American Chemical Society. Schematic of amorphization, reproduced with permission [18]. Copyright 2018, Wiley–VCH. Schematic of electrode surface, reproduced with permission [19]. Copyright 2019, Wiley–VCH. Schematic of cell, reproduced with permission [20]. Copyright 2020, American Chemical Society.

## 2. The Fundamentals

The ENRR is composed of multiple steps of proton-coupled electron transfers involving six protons and six electrons. The ENRR's half-reactions in acidic (1) and basic solutions (2) with standard reduction potential against reversible hydrogen evolution (RHE) reaction are given, respectively [14]

$$N_2(g) + 6H^+(aq) + 6e^- \rightleftharpoons 2NH_3(aq) \ (E^0 = 0.092 \ V_{RHE}) \tag{1}$$

$$N_2(g) + 6H_2O(l) + 6e^- \rightleftharpoons 2NH_3 + 6OH^-(aq) \ (E^0 = 0.092 \ V_{RHE}) \tag{2}$$

The Pourbaix diagram for $N_2$-$H_2O$ in Figure 1a represents thermodynamically favored species and phases depending on potential and the solution's pH [3]. Thermodynamically, only the ENRR is available and thus FE for $NH_3$ must be 100% at the potential and pH within the yellow-colored area. This area is between the red solid line representing $N_2$ reduction to $NH_4^+$ or $NH_3$ and the line ⓐ representing the hydrogen evolution's reaction (HER, $2H^+(aq) + 2e^- \rightleftharpoons H_2(aq)$, $E^0 = 0.000 \ V_{RHE}$). Furthermore, ENRR can be more favorable than HER at the potential below 0 $V_{RHE}$. However, the ENRR's performance is very poor in the yellow-colored area and the HER is more favorable than the ENRR at the potential below 0 $V_{RHE}$. The experimental observation can be supported by an additional challenge for the kinetic manner.

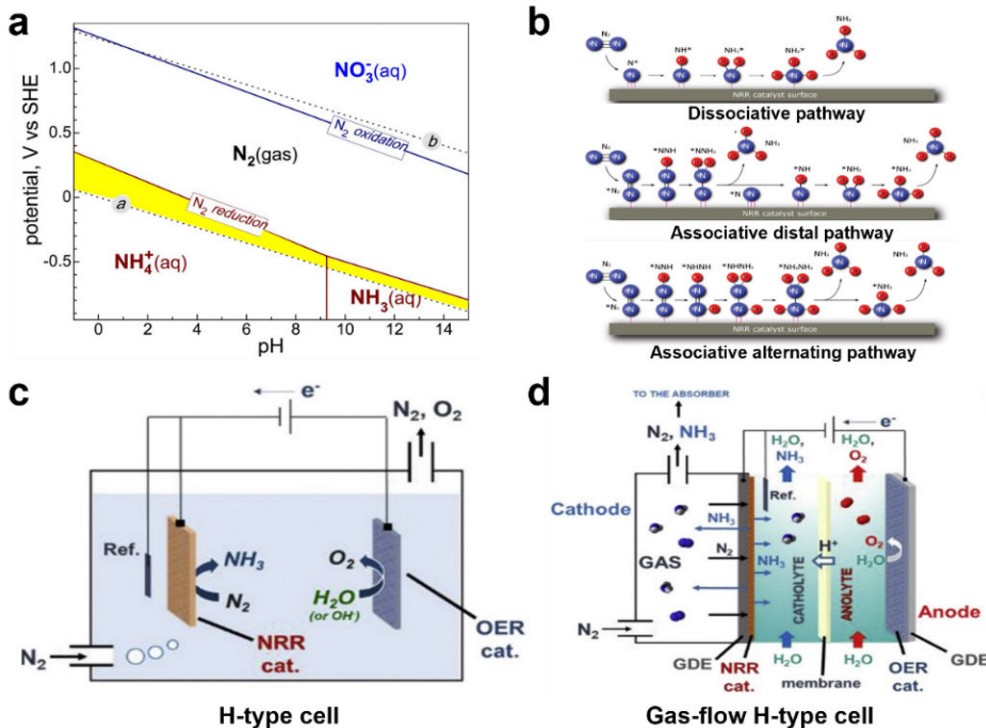

**Figure 1.** (**a**) Pourbaix diagram of nitrogen species, reproduced with permission [3]. Copyright 2018, The American Association for the Advancement of Science. (**b**) Schematic diagram of the $N_2$ reduction pathways, reproduced with permission [21]. Copyright 2019, Wiley–VCH. Reactor configurations of commonly used (**c**) H-type cell. Advanced cell configuration of (**d**) gas-flow H-type cell, reproduced with permission [22]. Copyright 2022, Elsevier B.V.

As shown in Figure 1b, pathways for $N_2$ to $NH_3$ conversion using six transferred electron and proton pairs are mainly represented as dissociative or associative steps [21]. In the dissociative pathway, the dissociated N atom is strongly adsorbed to the catalyst surface by the perfect cleavage of triple bonds from the $N_2$ molecule. Then, $NH_3$ is produced by multiple hydrogenations. However, this dissociative process is much more relatable to the $NH_3$ production via the Haber–Bosch process performed under high temperature

and pressure than the ENRR because the $N_2$ triple bonds cleavage requires enormous energy input. Meanwhile, the ENRR's mechanism is well-followed by the associative pathway where the first hydrogenation occurs with the adsorbed $N_2$ (*$N_2$), forming an *$N_2H$ intermediate [23]. The associative mechanism is further classified into two different pathways: the distal and alternating pathways. In the distal associative pathway, the N atom that is far from the catalyst's adsorption site is first hydrogenated with the three transferred electrons and protons and then released as $NH_3$. The other N atom still adsorbed on the catalyst conducts further hydrogenation to form $NH_3$. On the other hand, in the alternating pathway, the protons are alternately bonded with both N atoms until the final N-N bond is cleaved [14]. Finally, two $NH_3$ molecules are released. The initial step of $N_2$ adsorption and activation by forming the *$N_2H$ intermediate is a serious rate-determining step among the multiple steps for the ENRR. Furthermore, this multi-proton-coupled electron transfer process involves diverse intermediate formations, such as *NNH, *$NH_2$, and *NHNH with severe complexity. This results in sluggish kinetics relative to the HER, requiring a simple electron transfer with protons. Combining the thermodynamic and kinetic manner for the ENRR, a rational design of catalysts is critical for facilitating $N_2$ adsorption, activation, and selective $NH_3$ production with multiple byproduct evolution or competitive HER suppression.

The ENRR's performance is evaluated mainly using two factors of FE that represent the utilization ratio of the passed coulomb in $NH_3$ production and the production rate that shows the amount of $NH_3$ produced using a unit area (1 $cm^2$) of electrodes during an hour. The US Department of Energy (DOE) aims to achieve an FE of 50% for the ENRR and an $NH_3$ production rate of over 1700 $\mu g \ h^{-1} \ cm^{-2}$ [24]. However, the FE and production rate typically trade off in most reports [25,26]. Therefore, the suitable three-phase boundary of $N_2(g)$-catalyst(s)-$H_2O(l)$ with well-controlled mass transport of reactants and products, including catalyst engineering, is required to finally advance the $NH_3$ production rate [27]. The majority of ENRR has been performed in typical H-type cells, where the two compartments were separated with a membrane with a $N_2$ purging electrolyte as shown in Figure 1c [22,28,29]. However, since the $N_2$ gas solubility in $H_2O$ is significantly low, reaching as low as 0.0126 $mg \cdot g^{-1}$, the effective three-phase boundary representing active sites is insufficient, resulting in a low ENRR performance [20]. Thus, a strategy of adjusting the $N_2$ and $H_2O$ concentrations in catalysts needs to be suggested. For example, a proposed representative is promoting the $N_2$ concentration near the catalyst by directly supplying $N_2$ gas across a gas diffusion electrode (GDE) as shown in Figure 1d [22]. Like this strategy, diverse reaction environmental engineering is required to enhance performances. Considering the discussed ENRR fundamentals, the strategies to design catalysts and systems and their effect on the ENRR's performance will be reviewed in the next section.

## 3. Catalyst Engineering

Electrocatalysts for the ENRR have been developed using metals (Ag, Cu, W, Mo, Se, Fe, etc.), non-metals (Cl, B, F, P, S, O, N, etc.), metal oxides ($TiO_2$, $Bi_4V_2O_{11}$, etc.), metal sulfide ($MoS_2$), and carbon-based materials (graphdiyne, g-$C_3N_4$, N-doped carbon, etc.). In this section, we will review the representative strategies for ENRR improvement via facilitating $N_2$ activation and suppressing the competitive reaction.

### 3.1. Doping

Doping with atoms of different sizes and charge to the original electrocatalyst has been widely suggested in various electrocatalytic applications. The dopants can alter the electrocatalyst's electronic structure and change the binding strength of the reactant, intermediate, or product at active sites. Thus, they govern the electrocatalytic performance [30–33].

Recently, Qing Jiang et al. prepared Ag-doped Cu nanosheets cultivated on carbon paper (Ag-$Cu_{ns}$/CP) by a simple electrochemical deposition [34]. The prepared Ag-$Cu_{ns}$/CP exhibited a highly advanced ENRR compared to undoped $Cu_{ns}$/CP. For example, at

$-0.4$ $V_{RHE}$ in 0.1 M of $Na_2SO_4$, the Ag-$Cu_{ns}$/CP catalyst showed an FE of 20.9% and an $NH_3$ production rate of 4.56 µg h$^{-1}$ cm$^{-2}$, whereas $Cu_{ns}$/CP showed an FE of only 3.14% and an $NH_3$ yield rate of 2.28 µg h$^{-1}$ cm$^{-2}$. As revealed by X-ray photoelectron spectroscopy (XPS) on the Ag-doped Cu nanosheets sample, as shown in Figure 2a, the binding energy of $Cu^0$ positively shifted compared to the pristine Cu nanosheets and that of $Ag^0$ was negatively shifted compared to bulk Ag particles. This indicates that electrons from Cu sheets were partially transferred into Ag dopants, resulting in the formation of electron-deficient Cu sites in Ag-$Cu_{ns}$/CP. The electron-deficient Cu sites prevent proton adsorption and simultaneously provide opportunities to accept lone-pair electrons of $N_2$ molecules due to the suitable overlapping of orbitals from $N_2$ and electron-deficient Cu. Consequently, the properties induced by Ag doping on the Cu nanosheets suppress the HER and facilitate ENRR.

Carbon-based metal-free catalysts have been receiving great attention due to their attractive properties, such as high surface area, excellent conductivity, easily controllable defects, and effective cost [35]. To utilize these characteristics, Lele Duan et al. suggested metal-free Cl-doped ultrathin graphdiyne (Cl-GDY) electrocatalysts prepared by annealing the pristine GDY precursors with Ar and $Cl_2$ gases [36]. At $-0.45$ $V_{RHE}$ in 0.1 M HCl, the prepared Cl-GDY exhibited an FE of 8.7% and an $NH_3$ production rate of 10.7 µg h$^{-1}$ cm$^{-2}$, whereas pristine GDY exhibited an FE of 2.5% and a $NH_3$ production rate of 3.02 µg h$^{-1}$ cm$^{-2}$. Investigating the intensity ratio of the D to G band ($I_D/I_G$), representing the defect density and induced structural disordering of the carbon materials in Raman spectra revealed that the Cl-GDY exhibited a highly enhanced ratio compared with the pristine GDY. This implies that the Cl dopants increase defect sites, acting as active sites and structural distortions of GDY, resulting in improved ENRR performance.

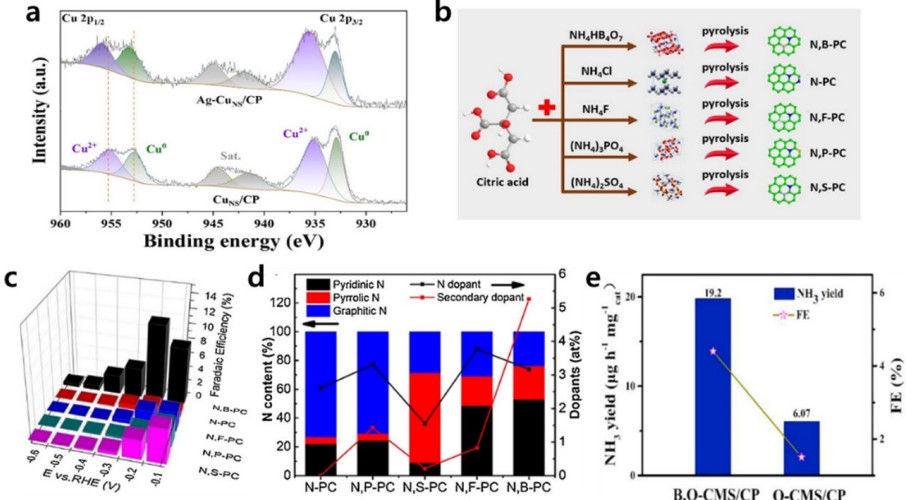

**Figure 2.** (**a**) Cu 2p XPS of Ag-$Cu_{NS}$/CP and $Cu_{NS}$/CP. Reproduced with permission [34]. Copyright 2021, Elsevier B.V. (**b**) Scheme of the heteroatoms co-doped porous carbons. (**c**) FEs of the heteroatoms co-doped porous carbons and N doped porous carbon. (**d**) Contents of total N and specific N species with secondary dopants. Reproduced with permission [37]. Copyright 2020, Elsevier B.V. (**e**) $NH_3$ production rate ($NH_3$ yield rates) and FEs of B,O-CMS/CP and O-CMS/CP. Reproduced with permission [15]. Copyright 2020, Elsevier B.V.

Furthermore, the synergistic effect of dual heteroatom doping for ENRR was reported using citric acid-derived carbon electrocatalysts. Zhong-Yong Yuan et al. prepared dual heteroatom doped porous carbon (PC) by combining N dopants with B, F, P, or S dopants as shown in Figure 2b [37]. Depending on the dopant's combination, their FEs were altered significantly as shown in Figure 2c. Among them, N, B-doped porous carbon (N,B-PC) exhibited the highest FE of 10.58% and an $NH_3$ production rate of 16.4 µg h$^{-1}$ cm$^{-2}$ at $-0.2$ $V_{RHE}$ in 0.1 M HCl. The major contribution of dual dopants was investigated

using Raman spectroscopy. Considering the ($I_D/I_G$), all dual-doped PCs, N,B-PC (1.025), N,F-PC (1.006), N,P-PC (0.978), and N,S-PC (0.976) demonstrated an increased structural disordering compared with the single N doped PC (0.962). Furthermore, N and B dual doping, pyridinic N ratio, and providing active sites for strong adsorption for $N_2$ largely increased compared to others in the XPS N 1s spectra (Figure 2d). Those two results indicate that dual heteroatom doping can effectively control the disordering and electronic structure of electrocatalysts affecting $N_2$ activation on active sites, resulting in ENRR improvement.

Dunmin Lin et al. also prepared dual-doped electrocatalysts, B,O-co-doped carbon microspheres (B,O-CMS) via a hydrothermal method [15]. The prepared B,O-CMS achieved a two-fold enhanced FE of 5.57% and an almost triple $NH_3$ production rate of 1.92 μg h$^{-1}$ cm$^{-2}$ compared with O-CMS at $-0.25$ V$_{RHE}$ in 0.1 M HCl as shown in Figure 2e. The B,O-CMS showed an increased $I_D/I_G$ ratio of 0.93 compared to the single doped O-CMS (0.88) in Raman spectra, indicating that the B,O-CMS has more defect sites acting as active sites than the O-CMS. Furthermore, as revealed by the Brunauer–Emmett–Teller (BET) measurements, the B,O-CMS provided more surface area than the O-CMS, in which each B,O-CMS and O-CMS showed a surface area of 488.7 and 373.6 m$^2$ g$^{-1}$, respectively.

These reports demonstrated that the doping strategy can adjust key factors for the ENRR by developing defect sites, providing or inducing neighbor's active sites, ordering/disordering the shift, surface area, and so on.

*3.2. Vacancy*

Surface vacancy formation is one of the most used strategies for creating active sites for ENRR. Vacant sites can have abundant localized electrons to preserve charge neutrality. The enriched electrons can weaken the $N_2$ triple bond, resulting in facilitating the ENRR [38]. In this part, we will thoroughly review the progress on oxygen ($O_v$), nitrogen ($N_v$), and sulfur vacancies ($S_v$) derived from various electrocatalysts ($TiO_2$, $C_3N_4$, $W_2N_3$, and $MoS_2$).

The $O_v$ formation has been the most developed strategy for ENRR, particularly for metal oxide-based electrocatalysts. Zhenyu sun et al. prepared $O_v$ rich $TiO_2$ via a solvothermal method and then annealed at different temperatures of 700, 800, and 900 °C under Ar gas flow [39]. At $-0.12$ V$_{RHE}$ in 0.1 M HCl, the $O_v$ rich $TiO_2$ sample annealed at 800 °C achieved the highest FE of 6.5% and an $NH_3$ production rate of 3.6 μg h$^{-1}$ cm$^{-2}$ among others. For example, the pristine $TiO_2$ showed a low FE and $NH_3$ production rate (2.4% and 1.5 μg h$^{-1}$ cm$^{-2}$, respectively). They conducted O 1s XPS measurements to investigate atomic compositions and their chemical states on the catalyst's surface. Each $O_v$ rich $TiO_2$ sample, annealed at 700, 800, and 900 °C, was composed of $O_v$ 18.4, 32.2, and 25.6%, respectively. Furthermore, the electron localized near the $O_v$ sites managed to partially reduce the oxidation state of $Ti^{4+}$, which served as an active site for ENRR. As the annealing temperature increased from 700 to 800 °C, the $O_v$ amount also increased. However, when the annealing temperature increased from 800 to 900 °C, the $O_v$ amount decreased. This was mainly due to the unintentional crystal structure's change from original anatase to rutile over 800 °C. $O_v$ formation is much more favored in the anatase phase relative to rutile [40]. As expected, the amount of reduced $Ti^{4+}$ increased from 700 to 800 °C, but decreased from 800 to 900 °C in Ti 2p XPS. Those results reveal that reduced $Ti^{4+}$ sites coupled with $O_v$ were critical for the ENRR.

Jiantai Ma et al. prepared an $N_2$ vacant 2D $C_3N_4$ (2D $C_3N_4$-NV), showing good stability and a large specific surface area via annealing 2D $C_3N_4$ nanosheets at 600 °C under an $N_2$ gas flow for 15 min as shown in Figure 3a [41]. The prepared 2D $C_3N_4$-NV exhibited a highly advanced FE of 10.96% and an $NH_3$ production rate of 178.5 μg h$^{-1}$ cm$^{-2}$ at $-0.30$ V$_{RHE}$ in 0.1 M HCl compared with pristine 2D $C_3N_4$ in Figure 3b. The dramatic performance advances by $N_v$ formation were mainly due to the $N_2$ adsorption ability, which is regarded as the primary challenge for ENRR. Based on the $N_2$ adsorption–desorption isotherm measurements, the $N_2$ adsorption capacity of 2D $C_3N_4$-NV was 193.6 cm$^3$ g$^{-1}$, which is almost 2 times and 1.4 times better than that of bulk $C_3N_4$ (91.5 cm$^3$ g$^{-1}$) and 2D $C_3N_4$, respectively. Furthermore, density functional theory (DFT) calculations support

that electron-rich $N_v$ sites can facilitate $N_2$ adsorption via weakening the $N_2$ triple bonds, resulting in a superior ENRR performance for 2D $C_3N_4$-NV.

Shi-zhang Qiao et al. prepared $N_v$ on 2D layered $W_2N_3$ with excellent structural stability (NV-$W_2N_3$) by annealing $Na_2W_4O_{13}$ under an $NH_3$ gas flow [42]. The presence of $N_v$ sites on $W_2N_3$ was revealed via XPS N 1s spectra, where N vacancy peaks at 400.2 eV could be observed in Figure 3c. Furthermore, in the extended X-ray absorption fine structure (EXAFS), NV-$W_2N_3$ showed a weakened W-N bonding, indicating a decreased W-N bonding number compared to pristine $W_2N_3$ in Figure 3d. The pristine $W_2N_3$ showed an FE of 3.5% and an $NH_3$ production rate of 0.94 $\mu g\,h^{-1}\,cm^{-2}$ at $-0.2$ $V_{RHE}$ in 0.1 M KOH. The ENRR performance was significantly improved by $N_v$ formation, showing an FE of 11.67% and $NH_3$ production rate of 2.332 $\mu g\,h^{-1}\,cm^{-2}$ at the same experimental condition as shown in Figure 3e. From the DFT calculation, it is revealed that electron-deficient W in $W_2N_3$ induced by $N_v$ facilitates $N_2$ adsorption and lowers the thermodynamic limiting potential of ENRR.

Recently, strategies for inducing sulfur vacancies ($S_v$) have been widely proposed, such as $O_v$ and $N_v$ formation. The molybdenum disulfide ($MoS_2$), which is a representative electrocatalyst of HER, has been proposed as a catalyst for ENRR by $S_v$ engineering [43,44]. Deliang Chen et al. developed $S_v$-rich $MoS_2$ by molybdenite precursors annealing at 800 °C ($MoS_2$-800) in an Ar gas flow [45]. As a result, the prepared $MoS_2$-$S_v$ achieved an FE of 17.9% and an $NH_3$ production rate of 9.352 $\mu g\,h^{-1}\,cm^{-2}$ at $-0.35$ $V_{RHE}$ in 0.1 M HCl, showing twice and 3.5-times higher FE and $NH_3$ production rate for pristine $MoS_2$, respectively. The major contribution of $S_v$ was revealed in the XPS spectra of Mo 3d, where the peak attributed to $Mo^{4+}$ was positively shifted for $MoS_2$-800 compared with pristine $MoS_2$. On the other hand, in the XPS spectra of S 2p, the peaks attributing to $S^{2-}$ were negatively shifted for $MoS_2$-800 relative to $MoS_2$. The peak shift of Mo implies electron transferring from Mo to other defect sites, whereas the peak shift of S implies the existence of deficient electrons. Additionally, the $N_2$ temperature-programmed desorption ($N_2$-TPD) results clearly exhibit the critical effect of $S_v$ as shown in Figure 3e. The $MoS_2$-800 represents a much stronger $N_2$ physisorption peak at 98 °C and chemisorption peak at 385 °C compared to the pristine $MoS_2$, indicating that $S_v$ in $MoS_2$-800 enhanced the $N_2$ adsorption ability. The progresses discussed in this part shows that vacancy engineering can effectively alter oxidation states around vacancies, affecting $N_2$ adsorption by weakening the $N_2$ triple bonds and increasing $N_2$ adsorption capacities, consequently enhancing ENRR.

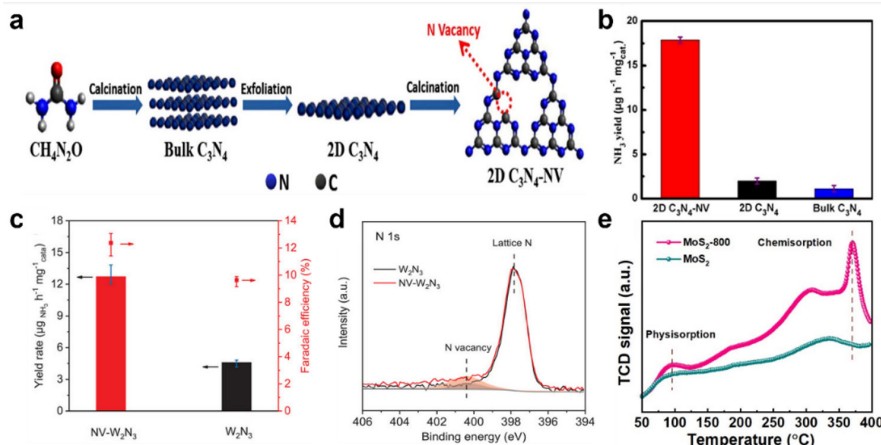

**Figure 3.** (**a**) Schematic diagram of 2D $C_3N_4$-NV synthetic route. (**b**) $NH_3$ production rates ($NH_3$ yield rates) of the 2D $C_3N_4$-NV, 2D $C_3N_4$, and bulk $C_3N_4$. Reproduced with permission [41]. Copyright 2021, Elsevier B.V. (**c**) $NH_3$ yield of NV-$W_2N_3$ and $W_2N_3$ at $-0.2$ $V_{RHE}$. (**d**) Synchrotron-based N 1s XPS of $W_2N_3$ and NV-$W_2N_3$. Reproduced with permission [42]. Copyright 2019, Wiley–VCH. (**e**) $N_2$-TPD curves of $MoS_2$ and $MoS_2$-800. Reproduced with permission [45]. Copyright 2021, Wiley–VCH.

### 3.3. Single Atom Catalysts

Single atom catalyst (SAC) synthesis has been regarded as a promising technique for catalyst development. The prepared SACs show outstanding catalytic performances with an extremely high number of active sites and unique electronic properties affecting catalytic pathways [17]. Metal SACs—such as Au, Cu, Fe, and so on—have been proposed and exhibited excellent ENRR performances.

Yongwen Tan et al. prepared Au SACs on nanoporous $MoSe_2$ ($Au_{SA}$/np-$MoSe_2$) by chemical vapor deposition followed by chemical etching [46]. They compared their ENRR performances with Au nanoparticles on np-$MoSe_2$ ($Au_{NPs}$/np-$MoSe_2$) and np-$MoSe_2$ alone as shown in Figure 4a,b. At $-0.3$ V$_{RHE}$ in 0.1 M $Na_2SO_4$, the $Au_{SA}$/np-$MoSe_2$ catalyst achieved an FE of 37.82% and an $NH_3$ production rate of 6.166 μg h$^{-1}$ cm$^{-2}$, whereas $Au_{NPs}$/np-$MoSe_2$ and np-$MoSe_2$ exhibited an FE and $NH_3$ production rate of 11.06% and 1.818 μg h$^{-1}$ cm$^{-2}$ and 11.78% and 0.69 μg h$^{-1}$ cm$^{-2}$, respectively. Since the Mo dichalcogenides including $MoSe_2$ provide excellent catalytic sites for HER [47], ENRR's representative competitive reaction, the major contribution for $NH_3$ production enhancement was expected for Au particles. From the XPS spectra of Au 4f, $Au_{SA}$/np-$MoSe_2$ showed a positively shifted peak attributed to $Au^0$ compared with $Au_{NPs}$/np-$MoSe_2$. This indicates that the Au single atom becomes electron-deficient due to a stronger electronic interaction between $Au_{SA}$ and $MoSe_2$, unlike the interaction between $Au_{NPs}$ and $MoSe_2$. Furthermore, Gibbs-free energies calculations for $N_2$ adsorption on Au additionally supported that the electron-deficient isolated Au single atoms on $MoSe_2$ efficiently inhibited the HER and facilitated $N_2$ adsorption with the following hydrogenation.

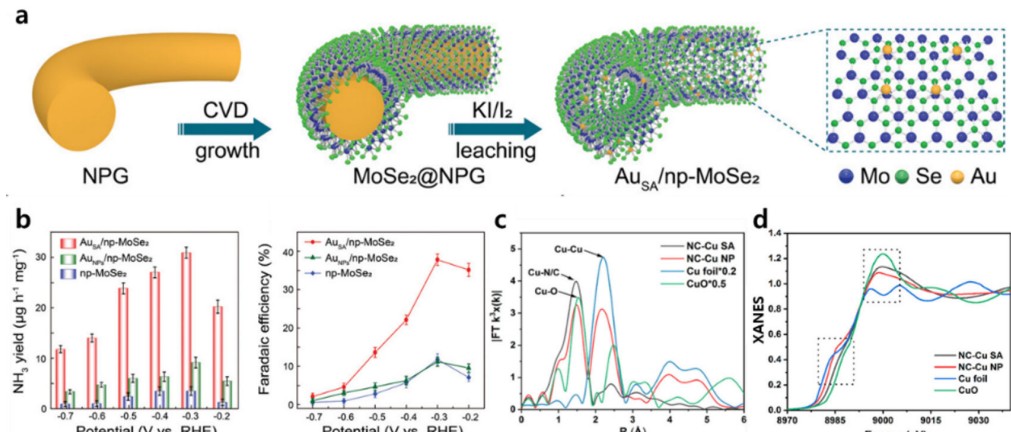

**Figure 4.** (**a**) Schematic illustration of $Au_{SA}$/np-$MoSe_2$ preparation process. (**b**) $NH_3$ production rates ($NH_3$ yield rates) and FEs of $Au_{SA}$/np-$MoSe_2$, $Au_{NPs}$/np-$MoSe_2$ and np-$MoSe_2$. Reproduced with permission [46]. Copyright 2021, Wiley–VCH. (**c**) EXAFS and (**d**) XANES spectra of NC-Cu SA, NC-Cu NP, Cu foil, and CuO. Reproduced with permission [48]. Copyright 2019, American Chemical Society.

Stephen J. Pennycook et al. synthesized Cu single atoms in porous N-doped carbon (NC-Cu SA) using a surfactant-assisted method [48]. As shown in Figure 4c, the presence of a single Cu atom was revealed by EXAFS for Cu K-edge. The NC-Cu SA represents the Cu-N/C peak with a negligible Cu-Cu peak, while NC-Cu NP shows only the Cu-Cu peaks representing the Cu nanoparticle. Furthermore, the single Cu atom in NC-Cu SA represents a positively charged valence state close to $Cu^{2+}$, unlike Cu particles representing a metallic $Cu^0$ state in NC-Cu NP as shown in Figure 4d. The electron-deficient Cu single atoms can provide strong $N_2$ adsorption sites as the first step for the ENRR. The NC-Cu SA catalyst exhibited relatively excellent performance with an FE of 13.8% and an $NH_3$ production rate of 53.3 μg h$^{-1}$ cm$^{-2}$ at $-0.35$ V$_{RHE}$ in 0.1 M KOH, whereas NC-Cu NP exhibited an FE of only 4.7% and an $NH_3$ production rate of 26.8 μg h$^{-1}$ cm$^{-2}$.

Naturally, $NH_3$ can be produced by the biological $N_2$ fixation over nitrogenase enzyme comprising Fe metal with ligands [49]. Xijun Liu et al. synthesized a Fe single atom on N-doped carbon frameworks ($Fe_{SA}$-NC) inspired by the nitrogenase using the hydrothermal method followed by carbonization [50]. The $Fe_{SA}$-NC exhibited an FE of 18.6% and an $NH_3$ production rate of 62.9 $\mu g\,h^{-1}\,cm^{-2}$ at $-0.4\,V_{RHE}$ in 0.1 M phosphate buffer solution. The ENRR performance of $Fe_{SA}$-NC was highly advanced compared to that of $Fe_{NPs}$-NC. The presence of single Fe atoms on NC was revealed by EXAFS, where the Fe-N bond was shown with a negligible Fe-Fe bond, representing the Fe atom's coordination with the nitrogen atoms but not between the Fe atoms. This result indicates that the Fe-N sites provide an active site for $N_2$ adsorption and that Fe single atoms contribute to weakening the triple-bond length of $N_2$ from 1.098 to 1.134 Å.

This subsection demonstrated that SACs have various properties compared to nanoparticle catalysts. For example, SACs change their oxidation state or those of surrounding materials, effectively adsorbing $N_2$ and weakening the triple bond, consequently enhancing the ENRR.

*3.4. Amorphization*

Surface amorphization is a promising catalyst engineering technique that partially or completely reduces the crystallinity of regular crystal structures on the catalyst's surface. Compared to highly crystalline structured catalysts, catalysts with reduced crystallinity or an amorphous surface nature can form a lot of defect sites comprising uncoordinated dangling bonds. Furthermore, they can provide lots of sites for reactant adsorption, which is typically the first sluggish step for catalytic reactions [51–53].

Qing jiang et al. prepared amorphous Au NPs on $CeO_x$-reduced graphite oxide (a-Au/$CeO_x$-RGO) using the co-reduction method [54]. The $CeO_x$ can reduce the crystallinity of Au and amorphized it. The RGO can provide substrates dispersing the Au NPs. The a-Au/$CeO_x$-RGO achieved an FE of 10.1% and an $NH_3$ production rate of 1.66 $\mu g\,h^{-1}\,cm^{-2}$ at $-0.2\,V_{RHE}$ in 0.1 M HCl, whereas crystalline Au/RGO only exhibited an FE of 3.67% and a $NH_3$ production rate of 0.7 $\mu g\,h^{-1}\,cm^{-2}$. In Au 4f XPS spectra, Au in both a-Au/$CeO_x$-RGO and c-Au/RGO shows a metallic state. However, in X-ray diffraction (XRD) patterns, unlike c-Au/RGO, a-Au/$CeO_x$-RGO shows no Au crystalline peak, indicating the amorphous state of Au/$CeO_x$-RGO. Thus, a-Au/$CeO_x$-RGO improves ENRR because its amorphous state has an abundance of unsaturated active sites.

Guihua Yu et al. prepared both amorphous and crystalline $Bi_4V_2O_{11}$/$CeO_2$ using electrospinning with a precursor solution containing different Ce:Bi ratios (1:2 for amorphous and 1:4 for crystalline) and followed calcination [55]. Since the $CeO_2$ adjusts heat transfer, affecting the neighbor metal's oxidation during calcination, the Ce:Bi ratio can control the crystallinity of $Bi_4V_2O_{11}$. The amorphous $Bi_4V_2O_{11}$/$CeO_2$ produces an FE of 10.16% and an $NH_3$ production rate of 46.42 $\mu g\,h^{-1}\,cm^{-2}$ at $-0.2\,V_{RHE}$ in 0.1 M HCl, which is about three-times higher FE and a 2.8-fold higher $NH_3$ production rate for crystalline $Bi_4V_2O_{11}$/$CeO_2$. The crystalline $Bi_4V_2O_{11}$/$CeO_2$ showed $Bi_4V_2O_{11}$ of orthorhombic type and $CeO_2$ cubic type in the XRD pattern, whereas amorphous $Bi_4V_2O_{11}$/$CeO_2$ showed no obvious $Bi_4V_2O_{11}$ peak. Interestingly, unlike the crystalline $Bi_4V_2O_{11}$/$CeO_2$ showing only $Bi^{3+}$ and $V^{5+}$, and dominant lattice O peaks in XPS spectra for Bi 4f, V 2p, and O 1s, respectively, the amorphous $Bi_4V_2O_{11}$/$CeO_2$ shows $Bi^{3+}$ and $Bi^{5+}$, $V^{5+}$ and $V^{4+}$, and an abundance of oxygen vacancy peaks. The additional $Bi^{5+}$ and $V^{4+}$ peaks with oxygen vacancies are observed only for the amorphous sample. This indicates that the amorphous $Bi_4V_2O_{11}$/$CeO_2$ have intrinsically localized electrons around the defect sites derived from amorphization. Furthermore, they can activate the $N_2$ adsorption and the first hydrogenation step using a transferred electron and proton.

Ke Chu et al. suggested amorphous $FeB_2$ nanosheets prepared by a facile reflux method as an electrocatalyst for ENRR [56]. This amorphous $FeB_2$ exhibited an FE of 16.7% at $-0.2\,V_{RHE}$ and an $NH_3$ production rate of 7.96 $\mu g\,h^{-1}\,cm^{-2}$ at $-0.3\,V_{RHE}$ in 0.5 M $LiClO_4$, which are 2- and 3-times higher FE and production rates for crystalline $FeB_2$, respectively.

This catalytic enhancement's major contribution was revealed by electrochemical active surface area (ECSA) measurements. The amorphous $FeB_2$ exhibited 1.56-times higher ECSA than crystalline $FeB_2$, indicating that amorphous $FeB_2$ has a large active surface area than the crystalline one. Additionally, considering the results of $N_2$ TPD, the amorphous $FeB_2$ showed a stronger chemical desorption peak than crystalline $FeB_2$, implying an enhanced $N_2$ adsorption ability of amorphous $FeB_2$ due to the increase in uncoordinated and defective sites in amorphous $FeB_2$.

These studies support the idea that amorphization is a suitable strategy for increasing the active surface area, number of active sites, effectively providing the appropriate $N_2$ adsorption strength.

The performance for ENRR depending on strategies for catalyst modification are summarized in Table 1.

**Table 1.** A brief summary of experimental studies on ENRR using various electrocatalytic engineering strategies.

| Catalyst Engineering | Catalyst | Electrolyte | Potential ($V_{RHE}$) | FE (%) | Production Rate ($\mu g\ h^{-1}\ cm^{-2}$) | Refs. |
|---|---|---|---|---|---|---|
| Doping | B,O-CMS | 0.1 M HCl | −0.25 | 5.57 | 1.92 | [15] |
| | S-CNS | 0.1 M $Na_2SO_4$ | −0.7 | 7.47 | 1.907 | [30] |
| | Mo-$MnO_2$ NFS | 0.1 M $Na_2SO_4$ | −0.4 | 12.1 | 3.46 | [33] |
| | Ag-$Cu_{ns}$/CP | 0.1 M $Na_2SO_4$ | −0.4 | 20.9 | 4.56 | [34] |
| | Cl-GDY | 0.1 M HCl | −0.45 | 8.7 | 10.7 | [36] |
| | N,B-PC | 0.1 M HCl | −0.2 | 10.58 | 16.4 | [37] |
| | Co/NC_500 | 0.1 M KOH | −0.1 | 10.1 | 10.2 | [57] |
| | PC/Sb/$SbPO_4$ | 0.1 M HCl | −0.15 | 31 | 3.34 | [58] |
| | PC/Sb/$SbPO_4$ | 0.1 M $Na_2SO_4$ | −0.1 | 34 | 2.7 | [58] |
| | Fe,Mo-N/C | 0.1 M $Na_2SO_4$ | −0.1 | 14.2 | 25.4 | [59] |
| Vacancy | CN/$C_{600}$ | 0.1 M HCl | −0.3 | 16.8 | 1.93 | [16] |
| | $O_v$ rich $TiO_2$ | 0.1 M HCl | −0.12 | 6.5 | 3.6 | [39] |
| | 2D $C_3N_4$-NV | 0.1 M HCl | −0.3 | 10.96 | 178.5 | [41] |
| | NV-$W_2N_3$ | 0.1 M KOH | −0.2 | 11.67 | 2.332 | [42] |
| | $MoS_2$-800 | 0.1 M HCl | −0.35 | 17.9 | 9.352 | [45] |
| | C@$CoFe_2O_{4-X}$ | 0.1 M $Na_2SO_4$ | −0.4 | 11.65 | 45.485 | [60] |
| | Ov-$TIO2$@C/Cu | 0.1 M $Na_2SO_4$ | −0.55 | 6.04 | 4.025 @ −0.6 $V_{RHE}$ | [61] |
| | Li-$TiO_2$(B) | 0.5 M $LiClO_4$ | −0.4 | 18.2 | 0.87 | [62] |
| | $CoS_{1-x}$ | 0.05 M $H_2SO_4$ | −0.15 | 16.5 ± 1.5% | 12.1 ± 1.4 | [63] |
| | $V_S$ -$MoS_2$ | 0.1 M $Na_2SO_4$ | −0.4 | 4.58 | 1.66 @ −0.5 $V_{RHE}$ | [64] |
| Single atom catalyst | $Au_{SA}$/np-$MoSe_2$ | 0.1 M $Na_2SO_4$ | −0.3 | 37.82 | 6.166 | [46] |
| | $Cu_{sa}$-NC | 0.1 M KOH | −0.35 | 13.8 | 53.3 | [48] |
| | $Fe_{sa}$-NC | 0.1 M phosphate buffer | −0.4 | 18.6 | 62.9 | [50] |
| | Ru SAs/N-C | 0.05 M $H_2SO_4$ | −0.2 | 29.6 | 30.8 | [65] |
| | $Y_1$/NC | 0.1 M HCl | −0.1 | 12.1 | 23.2 | [66] |
| | $Sc_1$/NC | 0.1 M HCl | −0.1 | 11.2 | 20.4 | [66] |
| | SA Ru-$Mo_2CT_X$ | 0.5 M $K_2SO_4$ | −0.3 | 25.77 | 12.17 | [67] |
| | W-NO/NC | 0.5 M $LiClO_4$ | −0.7 | 8.35 | 2.524 | [68] |
| | Mn–N–C SAC | 0.1 M NaOH | −0.45 | 32.02 | 5.36 @ −0.65 $V_{RHE}$ | [69] |
| | Pt SAs/$WO_3$ | 0.1 M $K_2SO_4$ | −0.2 | 31.1 | 171.2 | [70] |
| | AuSAs-NDPCs | 0.1 M HCl | −0.2 | 12.3 | 2.32 | [71] |
| Amorphization | a-Au/$CeO_x$-RGO | 0.1 M HCl | −0.2 | 10.1 | 1.66 | [54] |
| | amorphous $Bi_4V_2O_{11}$/$CeO_2$ | 0.1 M HCl | −0.2 | 10.16 | 46.42 | [55] |
| | amorphous $FeB_2$ nanosheets | 0.5 M $LiClO_4$ | −0.3 | 16.7 | 7.96 | [56] |

## 4. Three-Phase Boundary Engineering

The formation of the three-phase boundary (TPB) of $N_2$(g)-catalyst(s)-$H_2O$(l) is critical for the actual ENRR [27,72]. The advanced TPB can effectively suppress HER, the representative competitive reaction, and develop the $NH_3$ production rate. In this section, strategies for engineering TPB from electrode surface modification to systematic reaction environmental engineering will be discussed in detail.

### 4.1. Electrode Surface Modification

In typical aqueous ENRR systems, the low $N_2$ solubility ($0.0126$ mg·g$^{-1}$) prevents the effective transfer of $N_2$, unlike the free accessibility of $H_2O$, causing a significantly lower $N_2$ concentration near catalysts than $H_2O$ [20]. It provides an insufficient TPB condition, resulting in an ineffective FE and production rate for ENRR.

One strategy to advance the TPB is modifying the electrode's surface which has a partial hydrophobic nature. Xing Yi Ling et al. suggested coating over the Ag-Au catalyst with the hydrophobic zeolitic imidazolate framework-71 (ZIF-71) (Ag-Au@ZIF-71) by a wet chemical deposition [73]. The distinct characteristics of ZIF-71, having a unique pore structure comprising metal centers and hydrophobic functional groups of dichloroimidazole linkers provides a superhydrophobic barrier, effectively suppressing the free access of $H_2O$ to catalysts [74]. Furthermore, the $N_2$ molecules can freely diffuse through the ZIF-71 layer and thus be concentrated near the catalyst surface. The relatively reduced $H_2O$ concentration and increased $N_2$ concentration on catalysts by surface modification with ZIF-71 develop the TPB, effectively resulting in an enhanced ENRR. Ag-Au@ZIF-71 electrode exhibited an advanced FE of $18 \pm 4\%$ and an $NH_3$ production rate of ~$0.648$ µg h$^{-1}$ cm$^{-2}$ at $-2.9$ V$_{Ag/AgCl}$ in the solution of $0.2$ M LiCF$_3$SO$_3$ added to ethanol containing dry tetrahydrofuran (TPB), whereas the uncoated Ag-Au electrode showed an FE of only $9\%$ and an $NH_3$ production rate of $0.1296$ µg h$^{-1}$ cm$^{-2}$. Furthermore, continuous local $N_2$ saturation led to long-term stability with improved accessibility of $N_2$ to the catalyst's surface by ZIF-71. The Ag-Au@ZIF-71 represented constant CV features during 45 consecutive potential sweeps between $-3$ and $-0.5$ V$_{Ag/AgCl}$. On the other hand, the Ag-Au catalyst showed that the shape of CV was continuously deformed during the consecutive experiments, gradually flattening.

Miao Du et al. proposed a core–shell nanoporous gold (NPG) by encapsulating NPG with ZIF-8 (NPG@ZIF8) as shown in Figure 5a [19]. The 2-methylimidazole linkers comprising ZIF-8 provided a hydrophobic nature and suppressed $H_2O$ accessibility toward NPG catalysts [74]. As shown in Figure 5b, NPG@ZIF-8 showed an FE of $44\%$ and an $NH_3$ production rate of $22.0 \pm 0.3$ µg h$^{-1}$ cm$^{-2}$ at $-0.6$ V$_{RHE}$ in $0.1$ M Na$_2$SO$_4$, which is 7 times and a 10-fold higher FE and production rate than NPG alone, respectively. The improved ENRR performance mainly originated from ZIF-8 encapsulating NPG by effectively suppressing the competitive HER. NPG@ZIF-8 showed stability with very little change in FE and $NH_3$ production rate even after 10 consecutive experiments for 2 h each as shown in Figure 5c.

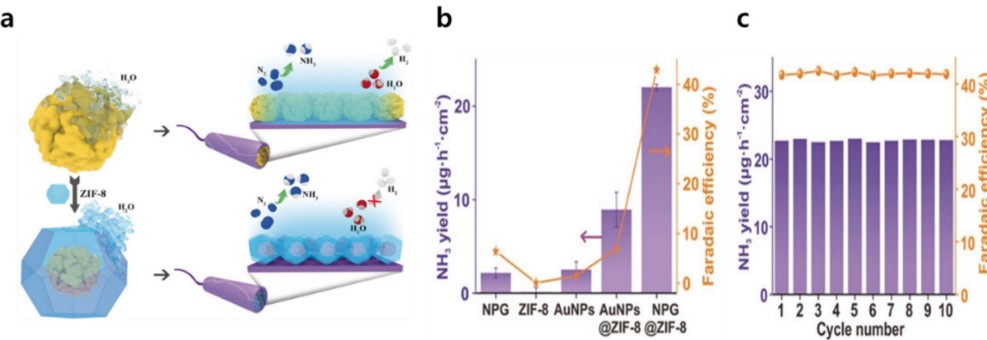

**Figure 5.** (**a**) Enhancement of $N_2$ reduction by the NPG@ZIF-8 electrocatalyst. (**b**) Comparison of the FEs and $NH_3$ production rates ($NH_3$ yield rates) for the presence and absence of ZIF-8 encapsulation at $-0.6$ V$_{RHE}$. (**c**) Stability tests of the NPG@ZIF-8 catalyst at $-0.6$ V$_{RHE}$. Reproduced with permission [19]. Copyright 2019, Wiley-VCH.

Even for photoelectrochemical applications, TPB engineering is important for producing $NH_3$ using $N_2$ and $H_2O$ with generated photoexcited electrons. Shuangyin Wang et al. proposed a surface modification of a Si-based photocathode using a hydrophobic PTFE porous framework [75]. The photo-excited electrons generated by light absorption

from Si were transferred to Au nanoparticles (Au NP), serving as active sites for ENRR. Since PTFE on Si photocathodes provides a hydrophobic environment originating from strong C-F bonds and resulting in low F polarization and low van der Waals force, $H_2O$ access was dramatically suppressed with accelerating $N_2$ diffusion toward Au NPs. As a result, the PTFE modified Si photocathode exhibited an FE of 37.8% and an $NH_3$ production rate 18.9 $\mu g\ h^{-1}\ cm^{-2}$ at $-0.2\ V_{RHE}$ in 0.05 M $H_2SO_4$ with 0.05 M $Na_2SO_3$ under 1 sun illumination. Its performance was 4- and 1.5-times higher than FE and production rate for pristine Si photocathode with Au NP, respectively.

Furthermore, Pei Kang Shen et al. suggested a way for accumulating $N_2$ molecules to increase the relative $N_2$ concentration, resulting in superior TPB for ENRR [76]. They prepared an aerobic-hydrophilic hetero-structured electrocatalyst using ultrathin $Bi_5O_7I$ nanotubes (UP-BOIN) and carbon spheres. The UP-BOIN served as the active sites and showed a highly porous surface structure with a diameter of about 5 nm with a hollow tubular geometry. This morphological trait provides a super-aerophilic nature. Then, the UP-BOIN was combined with the hydrophilic carbon sphere, modified by immersing them in $H_2O_2$ and 70 wt % $H_2SO_4$ in series. The combined aerophilic UP-BOIN and hydrophilic carbon sphere can control $N_2$ accumulation and $H_2O$ accessibility, affecting the superior TPB formation. For example, the 75% UP-BOIN with the 25% carbon sphere sample shows an FE of 6.10% and an $NH_3$ production rate of 2.286 $\mu g\ h^{-1}\ cm^{-2}$ at $-0.4\ V_{RHE}$ in 0.1 M $Na_2SO_4$. On the other hand, the electrode of 0% UP-BOIN with a 100% carbon sphere shows an FE and a production rate close to 0, indicating that UP-BOIN is a key electrocatalyst providing active sites. Furthermore, the sole UP-BOIN showed a significantly lower 5.19% FE and 0.796 $\mu g\ h^{-1}\ cm^{-2}$ $NH_3$ production rate compared to the carbon sphere combined UP-BOIN. This indicates that the effective control of reactant concentration, including $H_2O$, is critical for providing a sufficient TPB for the ENRR.

*4.2. Reaction Environmental Engineering*

As stated in previous sections, both catalyst modification and TPB advances can play key roles in ENRR development. Thus, it is significantly critical for developing superior TPB via controlling the local concentration of $N_2$ and $H_2O$ near catalysts. In this section, we will further discuss reaction environmental engineering strategies to develop TPB, including modification of $N_2$ gas supply, additives in electrolytes, etc.

Kyoung-Shin Choi et al. revealed how local $N_2$ and $H_2O$ concentrations influence ENRR by suggesting the simple cell modification for $N_2$ supplying [77]. As shown in Figure 6a, a BiSb alloy-based porous electrode—named the 'regular electrode'—was placed in the typically used setup, where ENRR was performed using dissolved $N_2$ molecules in $H_2O$. Additionally, a pseudo-gas diffuse electrode (pseudo-GDE) was proposed by changing the $N_2$ purging position into the electrode instead of the electrolyte, resulting in a direct $N_2$ gas supply near the catalyst as shown in Figure 6b. Since they compared the ENRR's performance depending on $N_2$ supply using perfectly consistent electrodes, the activity benefits from other possible activity-determining factors—such as electrode morphology, catalytic active sites, etc.—are negligible. As shown in Figure 6c, FEs from the pseudo-GDE were significantly altered against the $N_2$ flow rate because it could change the air pocket's size inside the electrode corresponding to the $N_2$ concentration. With the increase in the $N_2$ flow rate up to 100 mL $min^{-1}$, the FEs increased, representing the $N_2$ local concentration increase's positive influence on the ENRR. However, over the optimum rate, the efficiencies were decreased due to the significantly limited $H_2O$ supply for catalysts, resulting in reduced TPB sites. At $-0.6\ V_{RHE}$, with the optimal $N_2$ flow rate, the BiSb regular electrode showed an FE of 1.1% with an $NH_3$ production rate of 16.29 $\mu g\ h^{-1}\ cm^{-2}$ in 0.5 M borate. On the other hand, the BiSb pseudo-GDE exhibited 5% of FE, a five-fold improvement compared to the regular electrode. Furthermore, the electrode exhibited a highly advanced $NH_3$ production rate of 66.90 $\mu g\ h^{-1}\ cm^{-2}$, and the stability was shown through five experimental repetitions in which 5 C passed for each electrolysis.

Xiaofeng Feng et al. proposed a direct $N_2$ flow cell with a GDE which followed an additional hydrophobic layer modification [20]. Unlike the typically used H-cell which severely suffers from a low concentration of $N_2$ molecules near the catalyst due to low $N_2$ solubility (0.0126 mg·g$^{-1}$) in $H_2O$ as shown in Figure 6d, the flow cell with the gas-diffusion layer in Figure 6e can supply sufficient local $N_2$ concentration via direct humidified $N_2$ gas fed towards the cathode. Additionally, for the most effective TPB engineering, the developed electrode can be assembled with GDE, catalyst nanoparticles layer deposited on porous and hydrophobic carbon fiber paper, and hydrophobic layer in series as shown in Figure 6e. If the design is experimentally applied, the $N_2$ gas directly flows and diffuses. Additionally, the protons adjusted by passing the hydrophobic layer are also transferred towards the catalyst layer, which is expected to form a suitable TPB for ENRR by effectively suppressing HER.

Another interesting way to adjust $N_2$ concentration near catalysts is modifying the composition of electrolytes affecting $N_2$ solubility. For example, the non-aqueous ionic liquid (IL) can intrinsically inhibit HER because of the significantly low concentration of protons. For example, $N_2$ solubility is about 20-times higher in trihexyl(tetradecyl) phosphonium tris(pentafluoroethyl) trifluorophosphate, $[P_{6,6,6,14}][eFAP]$ (0.28 mg·g$^{-1}$), representative IL, than the solubility in $H_2O$ (0.0126 mg·g$^{-1}$) at room temperature [78]. Therefore, electrolyte engineering can give an attractive approach for controlling TPB for ENRR.

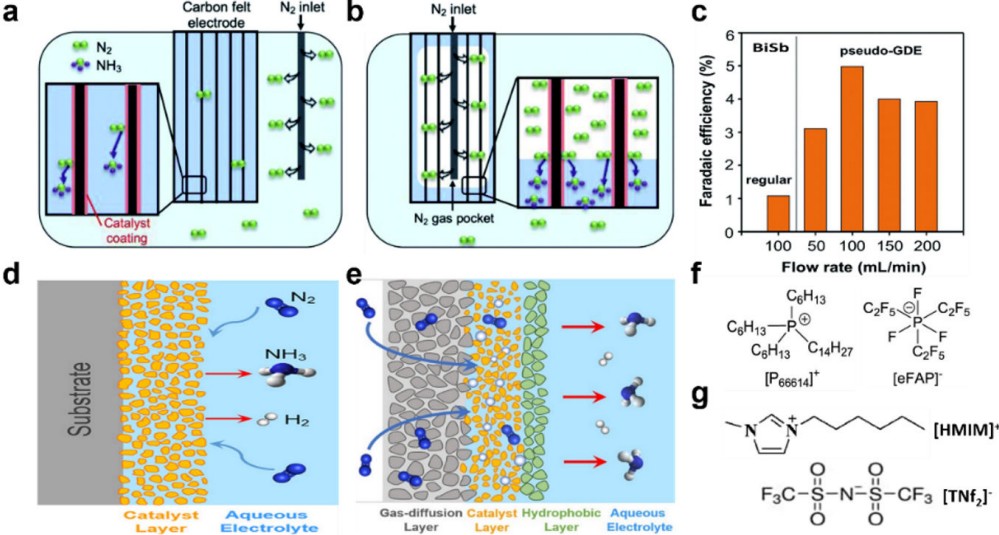

**Figure 6.** An illustration of (**a**) a regular electrode and (**b**) a pseudo-GDE. (**c**) FEs for $^{15}NH_3$ of BiSb as a regular electrode and as a pseudo-GDE with various $^{15}N_2$ flow rates at $-0.6$ V$_{RHE}$. Reproduced with permission [77]. Copyright 2021, Royal Society of Chemistry. Schematic illustration of two types of electrocatalytic interfaces for the ENRR. (**d**) Catalyst(s)-$H_2O$(l) interface in H-cell. (**e**) Proposed $N_2$(g)-catalyst(s)-$H_2O$(l) interface in a flow cell with GDE. Reproduced with permission [20]. Copyright 2020, American Chemical Society. Structure of (**f**) $[P_{6,6,6,14}][eFAP]$ Reproduced with permission [79]. Copyright 2018, American Chemical Society. (**g**) [HMIM][NTf$_2$]. Reproduced with permission [80]. Copyright 2007, American Chemical Society.

Douglas R. MacFalance et al. prepared a Fe electrocatalyst on fluorine-doped tin oxide glass (FTO) substrate by electro-deposition and ENRR was performed using the electrode in the electrolyte of $[P_{6,6,6,14}][eFAP]$ (Figure 6f) [81]. For comparison, ENRR was performed in another ionic liquid 1-hexyl-3-methylimidazolium bis(trifluoromethylsulfonyl)imide [HMIM][NTf$_2$] (Figure 6g) having a relatively low solubility of 0.017 mg·g$^{-1}$ than $[P_{6,6,6,14}][eFAP]$, showing 0.28 mg·g$^{-1}$ [80]. The differing $N_2$ solubilities in each electrolyte are due to the interaction between $N_2$ and [eFAP]$^-$ or [NTf$_2$]$^-$. According to the DFT calculation, the $N_2$ molecules can interact with [eFAP]$^-$ in two different ways: $N_2$ binding with the $C_2F_5$ group or with

F, showing a binding energy (Eb) of $-0.53$ and $-0.42$, respectively. On the other hand, $N_2$ molecules can make bonds with $CF_3$ groups of $[NTf_2]^-$ only showing an $E_b$ of $-0.16$. This indicates that $[eFAP]^-$ provides a much stronger $N_2$ binding strength than $[NTf_2]^-$, resulting in a more favorable $N_2$ solubility. This distinct $N_2$ solubility difference in the two ILs proportionally affects the ENRR. The Fe electrode exhibited an FE of $60 \pm 6\%$ in $[P_{6,6,6,14}][eFAP]$, whereas it showed an FE of 0.64% in $[HMIM][NTf_2]$ at $-0.8$ $V_{NHE}$.

Zhongfang Chen et al. investigated polyethylene glycol (PEG) additive's effect in 0.05 M $H_2SO_4$ electrolyte for ENRR using $TiO_2$ nanoarray electrodes [82]. Even if the majority of the electrolyte is $H_2O$, a molecular crowding effect formed by hydrogen bonding between $H_2O$ and PEG can significantly suppress $H_2O$ movement. In contrast, since the $N_2$ molecule is non-polar and its movement was not restricted by the effect, the $N_2$ molecule can freely move in the electrolyte to reach the active site. As a result, the $TiO_2$ electrode's FE and $NH_3$ production rate in the electrolyte with PEG were recorded as 32.13% and 13.6 $\mu g\ h^{-1}\ cm^{-2}$ for $-0.3$ $V_{RHE}$. On the other hand, it demonstrated an FE and production rate of only 0.8% and 5.1 $\mu g\ h^{-1}\ cm^{-2}$ without PEG.

In case sufficient protons are present near the catalytic active site, $H_2$ can be easily adsorbed on electrode surfaces before $N_2$, leading to a poor ENRR performance. Meanwhile, local $H_2O$ accessibility and its adsorption on active electrode surfaces can be controlled by developing the fundamentals of the electrical double layer (EDL) model [83,84]. For example, if we add additional alkali metal cations to the electrolyte, their hydration is promoted in the beginning. It then suppresses $H_2O$ movement, finally reducing the local $H_2O$ concentration. Furthermore, $N_2$ absorption strength can be enhanced via the direct adsorption of cations on the surface electrode [85].

An-Xiang Yin et al. investigated the $K^+$ cation's effect in acidic potassium sulfate for ENRR using Bi catalysts supported on carbon black [85]. They found FEs for ENRR from 9.8% to 66% at $-0.6$ $V_{RHE}$ depending on the concentration of the $K^+$ from 0.2 to 1.0 M. Additionally, the $NH_3$ production rate increased from about 230 to 884 $\mu g\ h^{-1}\ cm^{-2}$. With the increase in $K^+$ concentration, the amount of hydrated cations increases. Thus, they can be concentrated in the diffusion layer near the surface at a low potential. The advanced performance is mainly due to proton movement suppression by the hydrated $K^+$ cations. Therefore, $N_2$ is first adsorbed to the electrocatalyst's surface and activated. Meanwhile, $K^+$ directly adsorbed on the electrode can change the Bi electrode's electronic structure, thereby reducing the Gibbs free energy difference for $N_2$ activation ($\Delta G_{*NNH}$). Thus, by adding the $K^+$ cation in electrolytes, direct proton adsorption was effectively suppressed and $N_2$ adsorption and activation was facilitated.

The performances for ENRR depending on strategies for environmental engineering are summarized in Table 2.

**Table 2.** A brief summary of experimental studies on ENRR using various TPB engineering strategies.

| TPB Engineering | Catalyst | Electrolyte | Potential ($V_{RHE}$) | FE (%) | Production Rate ($\mu g\ h^{-1}\ cm^{-2}$) | Refs. |
|---|---|---|---|---|---|---|
| Electrode surface | NPG@ZIF-8 | 0.1 M $Na_2SO_4$ | $-0.6$ | 44 | $22.0 \pm 0.3$ | [19] |
| | Ag-Au@ZIF-71 | 0.2 M $LiCF_3SO_3$ added to ethanol containing dry TPB | $-2.9$ $V_{Ag/AgCl}$ | $18 \pm 4$ | 0.648 | [73] |
| | AuNp@PTFE | 0.05 M $H_2SO_4$ with 0.05 M $Na_2SO_3$ | $-0.2$ | 37.8 | 18.9 | [75] |
| | UP-BOIN | 0.1 M $Na_2SO_4$ | $-0.4$ | 6.1 | 2.286 | [76] |
| Environmental engineering | BiSb(pseudo GDE) | 0.5 M borate | $-0.6$ | 5 | 66.9 | [77] |
| | Fe on FTO | $[P6,6,6,14][eFAP]$ | $-0.8$ $V_{NHE}$ | $60 \pm 6$ | 0.25 | [81] |
| | $TiO_2$ nanoarray | PEG in 0.05 M $H_2SO_4$ | $-0.3$ | 32.13 | 13.6 | [82] |

**Table 2.** *Cont.*

| TPB Engineering | Catalyst | Electrolyte | Potential (V$_{RHE}$) | FE (%) | Production Rate (µg h$^{-1}$ cm$^{-2}$) | Refs. |
|---|---|---|---|---|---|---|
| | nitrogen-doped CNS | 0.25 M solution of LiClO$_4$ | −1.2 | 11.5 | 96 | [83] |
| | Cu | 5 M LiClO$_4$ | −0.6 | 12.1 ± 0.8 | 12.06 ± 2.39 | [84] |
| | Bi on CB | acidic potassiumsulfate (pH 3.5, 1.0 mol L$^{-1}$ of K$^+$) | −0.6 | 66 | 884 | [85] |
| | Pt | gel polymer electrolyte and polyacrylic acid polymer with 6 M KOH | 0.5 V (two-electrode system) | 0.01% | 2.448 | [86] |
| | Nano-Fe/CFC | 1 M K$_3$PO$_4$ | 1.0 V~0.0 V | 16.68 | 4.83 | [87] |

## 5. Summary and Perspective

NH$_3$ is an essential precursor for chemicals that are widely used in a variety of industries. Furthermore, it is an important sustainable fuel and media for hydrogen storage. Electrochemical N$_2$ reduction for NH$_3$ production (ENRR) has been considered a greatly promising process alternative to the traditional Haber–Bosch process that requires a high-energy input and which emits enormous CO$_2$ gas. Since the ENRR is performed at an ambient temperature and pressure using N$_2$ gas and H$_2$O as a hydrogen resource, it is economically effective and environmentally friendly. Considering those ideal potentials, the scientific community has devoted considerable effort to advancing the performance metrics, including the FE and NH$_3$ production rate. This review presented ENRR's governing fundamentals, such as thermodynamics, mechanisms, and mass transport. We then introduced strategies for improving its performance.

To design robust electrocatalysts that could provide superior active sites for N$_2$ adsorption and activation while suppressing HER, defect engineering via heteroatom doping and vacancy formation have been suggested. Furthermore, effective new types of catalysts have been proposed, such as catalysts comprising a single atom grown on a supporter or having an amorphous phase on the surface. These strategies could mainly control electronic structures promoting N$_2$ adsorption, *N$_2$H, the first intermediate, formation, and further hydrogenation with inhibiting proton adsorption. Meanwhile, to advance the ENRR's performance, not only catalyst engineering but also promising TPB engineering have proved to be critical. Electrode modification with functional layers and reaction environmental modification using direct N$_2$ gas supply, ionic liquids, and additional cations have been proposed. The desired concentration of N$_2$ and H$_2$O near catalysts can be suggested from the modifications, overcoming the major challenges of N$_2$ mass transport caused by its low solubility in aqueous solutions, and relatively free H$_2$O accessibility. Recently, the research has also been proposed to improve performance by forming a compact and uniform solid electrolyte interphase (SEI) [88]. Via multiple engineering modifications, the Bi catalysts reported the highest ENRR performance with an FE of 66% and an NH$_3$ production rate of 884 µg h$^{-1}$ cm$^{-2}$ at −0.6 V$_{RHE}$ in acidic potassium sulfate (pH 3.5, 1.0 mol L$^{-1}$ of K$^+$).

However, although strategies discussed in this review significantly advanced the ENRR, the performance is still inadequate for the ultimate technical target of practical NH$_3$ production; namely an FE of 50% and a NH$_3$ production rate of at least 6120 µg h$^{-1}$ cm$^{-2}$ [12,13]. Therefore, a combination of strategies for catalysts and reaction engineering is necessary for developing ENRR with ultra-high performance. For example, electrochemical CO$_2$ reductions have similar challenges to ENRR, such as the facile competitive reaction of HER, sluggish first active intermediate formation using transferred electrons and protons, limited TPB caused by imbalanced CO$_2$, and robust H$_2$O concentration due to low CO$_2$ solubility [89–91]. However, their performance was significantly advanced, making them economically feasible by combining robust catalyst developments, environmental reactions,

and environmental engineering, including system and process modification. Benchmarking the signs of progress of $CO_2$ reduction proves that ENRR could be further improved.

To turn ENRR into a practically available pathway alternative to the Haber–Bosch process, the desired electrocatalysts, environmental conditions, and the other cell components must be sustainable for at least several months. However, those stability issues are barely investigated, even for catalysts tested only for a few hours. Electrocatalysts may decompose due to poisoning, dissolution, and deactivation of active sites [14]. Furthermore, proper reaction environmental conditions can gradually differ compared to the initial state due to varying reactant and product concentrations. Furthermore, the cell components may be decomposed due to unintentional corrosion, pressure changes, etc. Therefore, prolonged stability test periods are highly recommended because they are useful for figuring out the imperfect durability's major origins. If the remaining challenges are successfully solved using the new frontiers suggested above, the ENRR can boost the $NH_3$ production process, significantly reducing energy consumption and $CO_2$ emission by replacing the traditional Haber–Bosch process.

**Author Contributions:** Investigation, conceptualization, and categorization by Y.H.M., N.Y.K. and S.M.K.; Writing and editing by Y.H.M. and N.Y.K.; Funding acquisition, editing, and supervision by Y.J.J. This manuscript was written through contributions of all authors. All authors have read and agreed to the published version of the manuscript.

**Funding:** This research was funded by the Graduate school of Post Plastic specialization of Korea Environmental Industry & Technology Institute grant funded by the Ministry of Environment of Republic of Korea, the National Research of Korea (NRF) grant (no. 2021R1F1A1063146), and the Korea Institute of Energy Technology Evaluation and Planning (KETEP) and the Ministry of Trade, Industry & Energy (MOTIE) of the Republic of Korea (no. 20202020800330).

**Data Availability Statement:** The datasets used and analyzed during the current study are available from the corresponding references listed.

**Conflicts of Interest:** The authors declare no conflict of interest.

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
