# Peer review of "Recent Advances in Electrochemical Nitrogen Reduction Reaction to Ammonia from the Catalyst to the System"

_catalysts, doi:10.3390/catal12091015_

Round 1
Reviewer 1 Report
In this review Jang and co-workers introduce the fundamentals of electrochemical nitrogen reduction reaction, elaborate the catalysts engineering including doping, vacancy inducing, single atom catalysts as well as amorphization, introduce the catalytic system improvements, and outlook the further direction of developments. The paper is well organized, exhaustive and comprehensive, and may serve as a competent guide for researchers to rapidly learn what has been done and what is yet to be done in this field. As electrochemical nitrogen reduction reaction to ammonia is an emerging and important scientific topic, I am supportive of its publication. Here are some suggestions for improving this review.
1. The conventional ammonia generation process is indeed energy-intensive and emits large amount of CO2. But most CO2 emission is caused by the hydrogen generation from methane. From this aspect, electrochemical generation of hydrogen is the key to solve the problem, rather than N2 reduction into ammonia, especially in consideration of the easiness of electrochemical hydrogen evolution reaction. Thus, is it a better way to generate ammonia by hydrogen generation from water electrolysis and then ammonia generation from hydrogen and nitrogen? I would highly appreciate if the authors could prove quantitative analysis of the energy consumption and CO2 emission of 1. conventional ammonia generation process, 2. direct electrochemical nitrogen reduction reaction for ammonia generation, and 3. the indirect process I mentioned above. I believe this quantitative analysis help me and other readers understand the significance of ENRR.
2. Aqueous ENRR systems (H-cell) are totally different from flow cells with GDE, as the catalysts in H-cells are immersed in aqueous while the catalyst in flow cell face both electrolyte and gas supply. Taking oxygen reduction reaction as an example, almost all high catalytic performance of ORR in H-cell cannot be transferred into a practical fuel cell with GDE. This may imply us the same potential situation in ENRR. As the authors said, the solubility of N2 prevents the ENRR efficiency in typical aqueous ENRR systems. It seems that as the exist of such a limitation, aqueous ENRR systems (H-cell) have zero possibility of being practical electrolysis system with large scale. If we accept this, the catalytic testing in such system seems meaningless. How do the authors comment on this?
3. Could the author provide more outlook on the further direction of potential development of electrolyzer?
4. One new important research should be added. Electrosynthesis of ammonia with high selectivity and high rates via engineering of the solid-electrolyte interphase, by Ib Chorkendorff et.al.
Author Response
Thank you for giving us valuable comments.
We carefully responded to your comments and revised the manuscript.

Reviewer 2 Report
Comments to the authors
The review article reports recent advances adopted for the electrochemical nitrogen reduction reaction to Ammonia. Designing an efficient catalytic system for electrochemical reduction of nitrogen to Ammonia have received increased attention in the last few decades. The present work affords a promising strategy for consolidating all the strategies designed for effective and efficient electrochemical reduction of nitrogen to Ammonia. I suggest minor revision for quality enhancement, which are given bellow.
11. The manuscript contains numerous grammatical and typographical mistakes which need correction and the language need sufficient improvement.
22. The first paragraph of Figure 1b discussion need references.
33. How your review article is different from the published articles. Make a table of previous published review article on ENRR and prove superiority of your current article over that articles.
44. Mention some advanced industries using ammonia in the introduction section.
55. Full stop should be always after the references like [12].
66. Figure 4b is not discussed in the article text. Why?
77. Authors are suggested to study the effect of coupling of different metal and nonmetals on the ENRR activity.
88. Table 1 and table 2 are not discussed nor mentioned in the main manuscript. Why?
Author Response
Thank you for your valuable comments.
We carefully responded to your comments and revised the manuscript.
